# Study on the Vibration Variation of Rock Slope Based on Numerical Simulation and Fitting Analysis

**Bing Yan [1], Ming Liu [1], Qingsheng Meng [2,\*], Yao Li [3], Shenggui Deng [1] and Tao Liu [2,4,\*]**

1 College of Marine Geosciences, Ocean University of China, Qingdao 266100, China; yanbing@stu.ouc.edu.cn (B.Y.); mingliuouc@163.com (M.L.); dengsg@ouc.edu.cn (S.D.)
2 College of Environmental Science and Engineering, Ocean University of China, Qingdao 266100, China
3 SGIDI Engineering Consulting (Group) Co., Ltd., Qingdao Branch, Qingdao 266011, China; yanbinggraduate@163.com
4 Laboratory for Marine Geology, Pilot National Laboratory for Marine Science and Technology, Qingdao 266061, China
\* Correspondence: qingsheng@ouc.edu.cn (Q.M.); ltmilan@ouc.edu.cn (T.L.)

**Abstract:** In engineering blasting, the slope surfaces in the blasting area exert various effects on the blast vibration velocity. For example, the slope effect and the whipping effect are generated in the slope area, which will influence the blast vibration velocity. The slope area is the key protection area for many projects; therefore, it is of practical value to explore the influence of slope surface on blast vibration speed for the prediction of blast vibration and protection against it. The influence of slope effect and whipping effect on blast vibration velocity in the slope area was analyzed by numerical simulation and fitting. The field monitoring data were fitted to the blast vibration velocity prediction formula. According to the obtained fitting formula, we inferred that vibration speed amplification occurred in the slope area. Numerical simulation was carried out using the ANSYS/LS-DYNA program. Using the above two methods, whether the slope effect and whip tip effect occurred in the study area was verified. By numerical simulation, we established three-dimensional (3D) slope models for four different working conditions. We simulated the complete blasting process and the consistency between the simulation results, and the field data proved the reliability of the numerical simulation. Based on the results of the numerical simulation, we explored the variation of blasting vibration velocity under different height difference conditions. Finally, we explored the distribution law of blasting vibration at the slope surface and inside the slope.

**Keywords:** blasting; fitting analysis; numerical simulation; blasting vibration velocity; whipping effect; slope effect

## 1. Introduction

Blasting methods are commonly used in engineering fields such as pit excavation, mining, and tunnel construction. The blasting in the rock and soil will form elastic waves and cause the vibration of the adjacent strata [1,2]. Blasting vibration, if not controlled, can have an adverse effect on the surrounding buildings [3]. Therefore, blasting tests are needed before blasting. Researchers have fitted the test data to the prediction equation to obtain a more accurate blast vibration prediction equation [4,5].

The effects of blasting vibration on rocky slopes have been extensively investigated. Yu et al. [6], Tan et al. [7], and Jiang et al. [8] studied the changes of blast vibration in the slope area in mine and reservoir projects and summarized the changes of blast velocity inside the slope. Jiang et al. [9] evaluated the influence of underground mining on the stability of slopes in open pit mines and established a blast vibration prediction formula through theoretical analysis. Zhang et al. [10] analyzed the effect of internal elevation of slopes on blasting vibration, derived a blast vibration prediction equation, and verified the accuracy of the equation by comparing the calculation results with the field data. However,

these studies were mostly conducted by combining field monitoring and theoretical analysis. With the development of computer technology, numerical simulation software has been widely applied to the study of slope blasting vibration distribution law. Xu et al. [11] used ANSYS/LS-DYNA software to simulate the whole process of blast vibration propagation in the slope area of the mine site and summarized the stress state and deformation in the slope area. Yang et al. [12] used FLAC 3D software to simulate the dynamic process of the slope under the influence of blasting load. Roslan et al. [13] used the UDEC method to simulate the stress wave propagation inside the slope and the change law during blasting. Bazzi et al. [14] investigated the effect of blast vibration on slope stability using finite element analysis. In summary, studies on blast vibration in slope areas can be mainly conducted with three analysis methods, namely, field monitoring, theoretical analysis, and numerical simulation. The amplification of blast vibration inside the slope can be explained by the whipping effect [15] and the slope effect [16]. Wang et al. [17], Ji et al. [18], and Ye et al. [19] demonstrated the existence of the whip effect at slopes and steps. Li et al. [20], Yang et al. [21], and Cai et al. [22] analyzed the influence of the internal slope effect on blast vibration velocity.

In this paper, the distribution law of blasting vibration inside the slope was studied. Firstly, we monitored the blasting vibration at the site and obtained a blasting vibration prediction formula applicable to the geological conditions at the site by fitting analysis. According to the blasting vibration prediction formula, we preliminarily concluded that the amplification effect on vibration speed occurred inside the slope. Secondly, we established a three-dimensional model of the actual slope by means of numerical simulation. We compared the field monitoring data with the numerical simulation results to prove the reliability of the simulation results. Finally, in order to obtain the effect of slope angles on blast vibration speed, we set up four working conditions. Based on the numerical simulation results, we investigated the distribution law of blast vibration at the slope surface and inside the slope.

## 2. Project Summary

We planned to build a foundation pit for a hospital project in phase II with a construction land area of about 41,000 square meters. The circumference of the foundation pit was about 800 m, and the depth of the foundation pit was 15–49 m. Hospitals, residential areas, and other buildings were located around the foundation pit.

In order to investigate the change law of blasting vibration inside the slope, we conducted on-site monitoring, and the blasting vibration prediction formula was fitted according to the monitoring data. The monitoring instrument was the TC-6850N blast vibration collector. We set the monitoring points in the flat area at the top of the slope. The monitoring distribution points are shown in Figure 1.

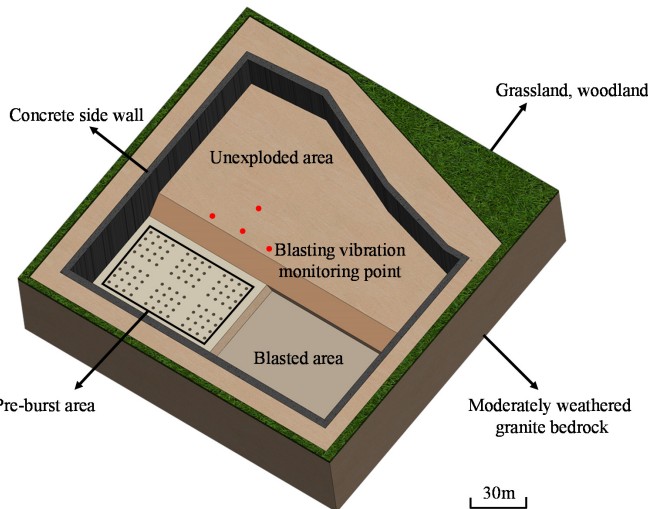

**Figure 1.** The schematic diagram of the project site and monitoring points.

### 3. Fitting Analysis of Blasting Vibration Prediction Formula Considering Elevation Value

The prediction data of Sadov's formula and previous research show that blasting vibration velocity will decrease with the increase of blasting distance [23]. However, a large number of foundation pits, slopes, and mine projects show that the vibration velocity will be amplified to a certain extent with the increase of elevation difference. In this regard, monitoring points were set up in the excavation blasting construction site to explore the response law of the foundation pit slope to blasting vibration during the blasting excavation at the bottom of the foundation pit.

The existing formula reflecting the influence of elevation on blasting vibration velocity is as follows [24]:

$$V = K \left( \frac{Q^{\frac{1}{3}}}{D} \right)^{\alpha} \left( \frac{Q^{\frac{1}{3}}}{H} \right)^{\beta} \tag{1}$$

where $K$ and $\alpha$ are the coefficient and attenuation index related to the topographic and geological conditions between the bursting points and the calculation of the protected object; $Q$ is the maximum single section charge of blasting, kg; $V$ is the designed vibration velocity for the ground particle where the protected object is located, cm/s; $D$ is the horizontal distance from the bursting point to the measuring point, m; $H$ is the height difference between the bursting point and the measuring point, m; $\beta$ is the elevation coefficient.

Equation (1) introduces new variables, height difference ($H$) and horizontal distance ($D$), and is a variation of Sadov's formula. It is only applicable to situations where there is a certain elevation difference between the burst center and the target mass point (the measurement point is higher than the burst center).

Generally speaking, a smaller elevation coefficient $\beta$ indicates a greater influence of elevation and a more significant amplification effect [25,26]. Therefore, by fitting the blast vibration prediction formula, we can make a preliminary inference that the blast vibration is affected by the elevation.

The TC-6850N blast vibration collector can collect blast vibration data in real time at a specific location in the X (vertical), Y (horizontal), and Z (tangential) directions. The horizontal distance ($D$) and height difference ($H$) need to be determined prior to blast vibration data collection. After being acquired, the data can be summarized in the form of an electronic file. It is important to note that the resultant velocity is not directly generated by the data collector but is calculated after data compilation. Therefore, the vertical and horizontal blast vibration velocities were selected for analysis. On-site monitoring data of vertical and horizontal vibration velocities were taken as fitting data, as shown in Table 1.

**Table 1.** The data collected in the field.

| Weight of Explosive (kg) | Number of Survey Point | Horizontal Distance $D$ (m) | Height Difference $H$ (m) | Vertical Vibration Velocity $V_V$ (cm·s$^{-1}$) | Horizontal Vibration Velocity $V_L$ (cm·s$^{-1}$) |
|---|---|---|---|---|---|
| 3.6 | | 6.94 | 15.85 | 4.476 | 3.240 |
| 3.6 | 1 | 13.42 | 16.25 | 0.957 | 0.857 |
| 3.6 | | 21.37 | 16.25 | 0.368 | 0.335 |
| 3.6 | | 13.29 | 15.85 | 0.989 | 0.869 |
| 3.6 | 2 | 8.29 | 16.25 | 3.672 | 2.982 |
| 3.6 | | 16.4 | 16.25 | 0.921 | 0.836 |
| 3.6 | | 18.908 | 15.55 | 0.648 | 0.617 |
| 3.6 | 3 | 15.26 | 15.85 | 0.765 | 0.821 |
| 3.6 | | 17.29 | 16.59 | 0.688 | 0.701 |

According to the site monitoring data, the blasting vibration prediction formula was modified to obtain a new formula suitable for the geological conditions of the site. The blast velocity prediction equation in the vertical direction ($V_V$), considering the elevation value, is shown below:

$$V_V = 68.03 \left( \frac{Q^{\frac{1}{3}}}{D} \right)^{2.14} \left( \frac{Q^{\frac{1}{3}}}{H} \right)^{-0.23} \tag{2}$$

The prediction equation for blast vibration speed in the horizontal direction ($V_L$), considering the elevation value, is shown below:

$$V_L = 56.26 \left( \frac{Q^{\frac{1}{3}}}{D} \right)^{1.939} \left( \frac{Q^{\frac{1}{3}}}{H} \right)^{-0.085} \tag{3}$$

By fitting data, Equations (1)–(3) were obtained. The following conclusions can be obtained by comparing Equations (2) and (3).

The correlation coefficients ($R^2$) of Equations (2) and (3) are 0.946 and 0.93, indicating that there is a strong correlation between the fitting results and the field monitoring data and the research error is small.

In Equation (2), the elevation coefficient is $\beta = -0.23$, and in Equation (3), the elevation coefficient is $\beta = -0.085$. By comparing the research results in references [7,9,25,26], it can be learned that the vertical vibration velocity is more significantly influenced by the elevation than the horizontal vibration speed and that the blast vibration amplification effect increases with the rise in elevation.

It can be determined based on the type of formula that the blast vibration speed decreases with the increase of horizontal distance ($D$) under the case of fixed height difference ($H$). With the depletion of energy in the blast vibration wave propagating to the distance, the vibration speed will also tend to be 0.

The error rate between the measured value and the predicted value of the regression formula was analyzed to verify the accuracy of the fitting formula. The error rate results are shown in Table 2.

**Table 2.** The error rate statistics of the prediction formula.

| Formula Fitting Results | Type of Vibration Velocity | Maximum Error Rate (%) | Average Error Rate (%) |
|---|---|---|---|
| $V = 68.03 \left( \frac{Q^{\frac{1}{3}}}{D} \right)^{2.14} \left( \frac{Q^{\frac{1}{3}}}{H} \right)^{-0.23}$ | Vertical | 20.42% | 12.97% |
| $V = 56.26 \left( \frac{Q^{\frac{1}{3}}}{D} \right)^{1.94} \left( \frac{Q^{\frac{1}{3}}}{H} \right)^{-0.085}$ | Horizontal | 23.98% | 14.95% |

As can be seen from Table 2, the maximum error rate of vibration velocity in the vertical direction was 20.42%, and the average value was 12.97%. The maximum error rate of vibration velocity in the horizontal direction was 23.98% and the average value was 14.95%. The average error rate calculated by the blast vibration prediction formula in reference [6] was 6% and 12%; the maximum error rate calculated by the blast vibration prediction formula in reference [27] was 4.8%. The error rate of the obtained blast vibration prediction formula was within an acceptable range considering the complex construction conditions at the site with uneven terrain and ubiquitous debris, and therefore it could be applied to the prediction of blast vibration at the site.

The traditional Sadov's formula was fitted, and the error rate was analyzed. The results are shown in Table 3.

The prediction error rate of Sadov's fitting formula was very large. The average error rate in the vertical direction reached 94.74%, and that in the horizontal direction reached 93.24%, indicating that Sadov's fitting formula was not suitable for the prediction of blast

vibration velocity of this project. The project should use the prediction formula, which considered the elevation value.

**Table 3.** The error rates of Sadov's fitting formula.

| Fitting Results of Traditional Sadov's Formula | Coefficient of Correlation | Type of Vibration Velocity | Average Error Rate (%) |
|---|---|---|---|
| $V = 0.0696\left(\dfrac{Q^{\frac{1}{3}}}{R}\right)^{0.165}$ | $R^2 = 0.902$ | Vertical | 94.74% |
| $V = 0.0709\left(\dfrac{Q^{\frac{1}{3}}}{R}\right)^{0.185}$ | $R^2 = 0.905$ | Horizontal | 93.24% |

## 4. Numerical Simulation of Slope Effect

Based on the construction conditions at the site, we conducted a numerical simulation using the ANSYS/LS-DYNA program. The ground investigation data showed that the surface plain fill was cleared and that the stratum was mainly composed of medium-weathered granite. We adopted the relevant parameters provided in the exploration data to define the physical properties of the medium-weathered granite. In the actual project, the slope height was 15 m, and the horizontal distance ($H_L$, from the foot to the top of the slope) was 6 m. In order to investigate the effect of slope angles on blasting vibration, we fixed the height of the slope in the 3D model to 15 m. We built four slope models with horizontal distances ($H_L$) of 6 m, 12 m, 18 m, and 24 m and slope angles of 68.20°, 51.34°, 39.80°, and 32°, respectively. The model for working condition II was divided into a total of 182,568 meshes. The fine meshes ensured the accuracy of the calculation and saved calculation time. The mesh quality and calculation accuracy are high enough for subsequent studies. The model is shown in Figure 2.

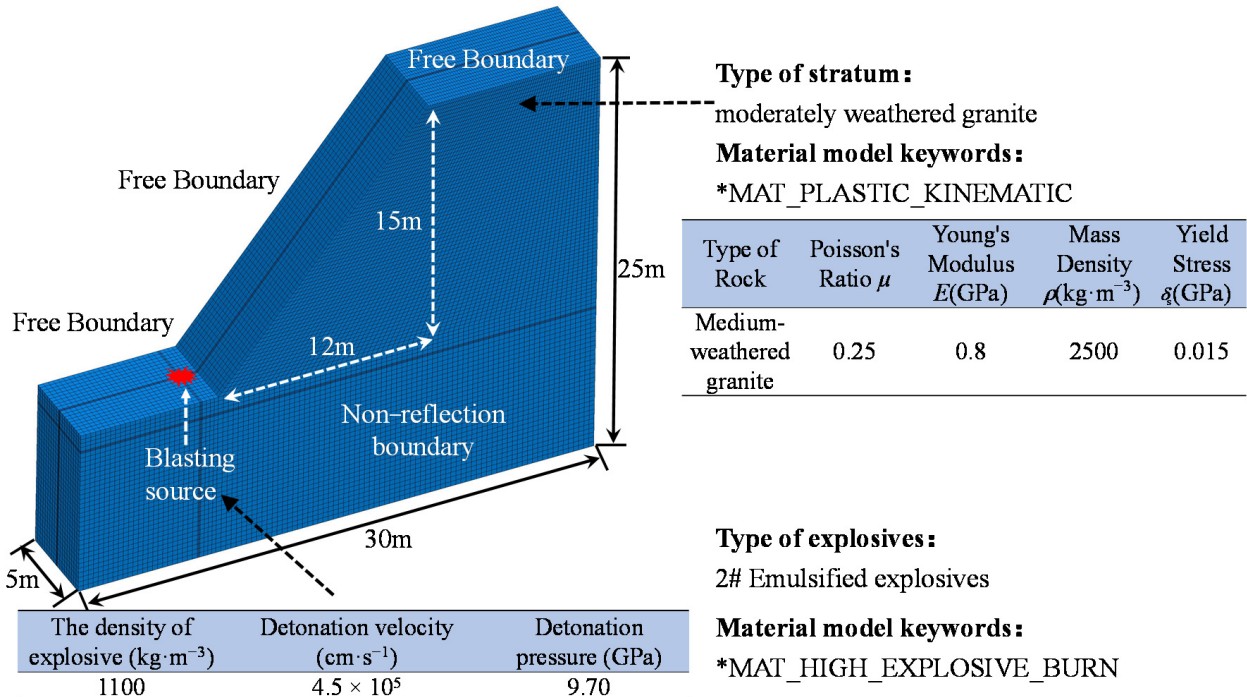

**Figure 2.** The 3D model of numerical simulation of work condition II.

For the constraint conditions, the top surface of the model was set as the free surface, and other surfaces were set as nonreflecting boundaries to simulate an infinite stratigraphic space. As for the load conditions, since this study focused on the vibration velocity variation of the

masses at the surface and the shallow stratigraphic interior, the effect of crustal stress was not considered. The detailed data for the four operating conditions are shown in Table 4.

**Table 4.** The detailed data of four working conditions.

| Working Condition | Weight of Explosive (kg) | Horizontal Blast Distance $H_L$ (m) | The Vertical Elevation Difference $H_V$ (m) | Slope Gradient (°) |
|---|---|---|---|---|
| Working condition I | | 6 | | 68.20 |
| Working condition II | 3.6 | 12 | 15 | 51.34 |
| Working condition III | | 18 | | 39.80 |
| Working condition IV | | 24 | | 32 |

The elastic–plastic material model was selected as the rock-soil mass model, and the keyword was *MAT_PLASTIC_KINEMATIC. Based on the site geological survey data, the selected mechanical parameters are shown in Table 5.

**Table 5.** The selection of rock stratigraphic parameters.

| Type of Rock | Poisson's Ratio $\mu$ | Young's Modulus $E$ (GPa) | Mass Density $\rho$ (kg·m$^{-3}$) | Yield Stress $\delta_s$ (GPa) |
|---|---|---|---|---|
| Medium-weathered granite | 0.25 | 0.8 | 2500 | 0.015 |

The explosive weight was 3.6 kg, and only one explosion was carried out. ANSYS/LS-DYNA software, which involves the numerical simulation of rock explosives, generally uses the 2 # emulsion explosive material model with the keyword *MAT_HIGH_EXPLOSIVE_BURN. This keyword must be defined with the equation of state; otherwise, there will be calculation errors. The equation of state keyword was *EOS_JWL, which contains the explosive-related internal energy parameter. In the case of the definition of the state equation, explosives in the explosion process, the explosion pressure, volume changes, and energy signal characteristics can be accurately described. The state equation is shown below [28]:

$$P = A\left(1 - \frac{\omega}{R_1 V}\right)e^{-R_1 V} + B\left(1 - \frac{\omega}{R_2 V}\right)e^{-R_2 V} + \frac{\omega E_0}{V} \tag{4}$$

where $P$ is the pressure; $V$ is the initial relative volume; $E_0$ is the internal energy constant; $A$, $B$, $R_1$, $R_2$, $\omega$ are the characteristic parameters, and for certain explosives, the value is constant.

Detailed parameters are shown in Table 6.

**Table 6.** The parameters of the explosive.

| The Density of Explosive (kg·m$^{-3}$) | Detonation Velocity (cm·s$^{-1}$) | Detonation Pressure (GPa) | JWL Equation of State Parameter | | | | | |
|---|---|---|---|---|---|---|---|---|
| | | | $A$ (GPa) | $B$ (GPa) | $R_1$ | $R_2$ | $\omega$ | $E_0$ (GPa) |
| 1100 | $4.5 \times 10^5$ | 9.70 | 214.4 | 0.182 | 4.2 | 0.9 | 0.15 | 4.19 |

*4.1. Comparison Analysis of Field Monitoring Data and Numerical Simulation Results*

The time-history change of blasting vibration at the top of the slope is shown in Figure 3. The numerical simulation result of the maximum vertical vibration velocity ($V_V$) at the top of the slope was 4.52 cm/s, and the monitoring value was 4.47 cm/s. The simulated maximum horizontal vibration velocity ($V_L$) was 3.06 cm/s, and the monitoring value was 3.24 cm/s. The numerical simulation results were close to the field monitoring results.

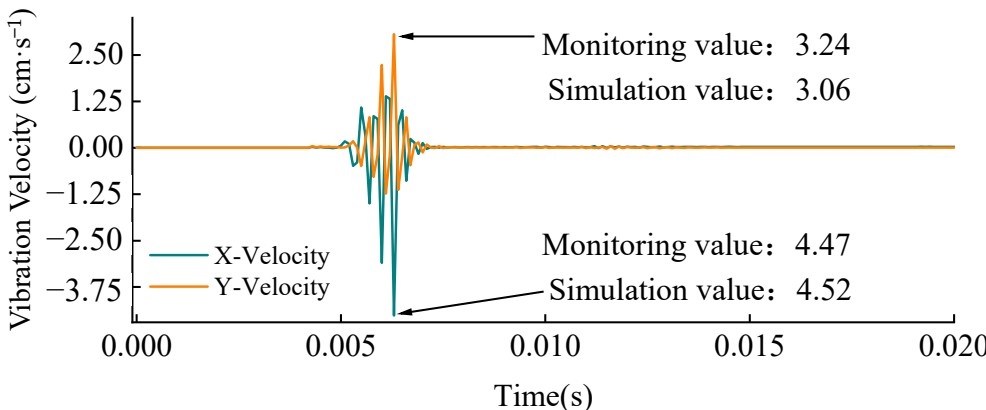

**Figure 3.** The change of blasting vibration duration at the top of the slope.

### 4.2. Effect of Height Values on Blasting Vibration

In this section, vertical and horizontal vibration velocities were selected for analysis. The effect of heights on blast vibration velocity within the same slope was analyzed, and the inference obtained by fitting Equation (2) and fitting Equation (3) was verified to be consistent with the numerical simulation results. In the slope model, feature points have the same horizontal distance ($D$) and various height differences ($H$) as the data from working condition I. The feature points were named $C_1$–$C_6$ from bottom to top, as shown in Figure 4.

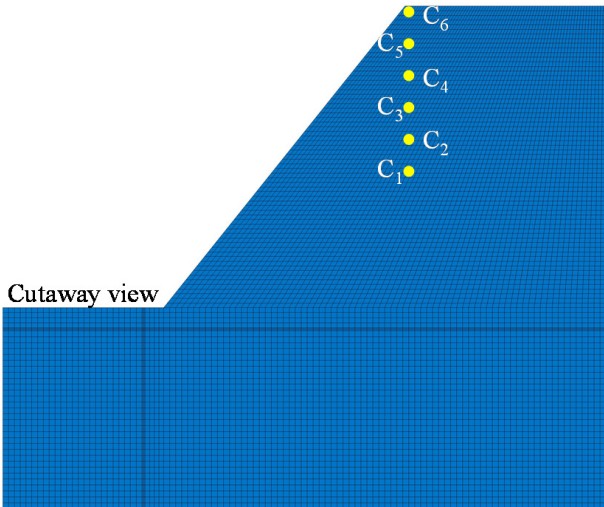

**Figure 4.** The feature points $C_1$–$C_6$.

The comparative analysis of vertical and horizontal vibration velocities is shown in Figure 5.

As can be seen from Figure 5, when the horizontal distance was constant, vibration velocities in both vertical and horizontal directions showed amplification near the top of the slope. The variation of vibration velocity in the vertical direction was more significant than that in the horizontal direction. In the vertical direction, the simulated minimum blasting vibration velocity was 2.44 cm/s, and the maximum value was 4.52 cm/s, with an increase of 85.24%. In the horizontal direction, the minimum blasting vibration velocity was 1.72 cm/s, and the maximum value was 3.06 cm/s. The increase rate was 77.91%, lower than that in the vertical direction, indicating that the vertical vibration velocity was more affected by the elevation, which is consistent with the law obtained by the fitting analysis.

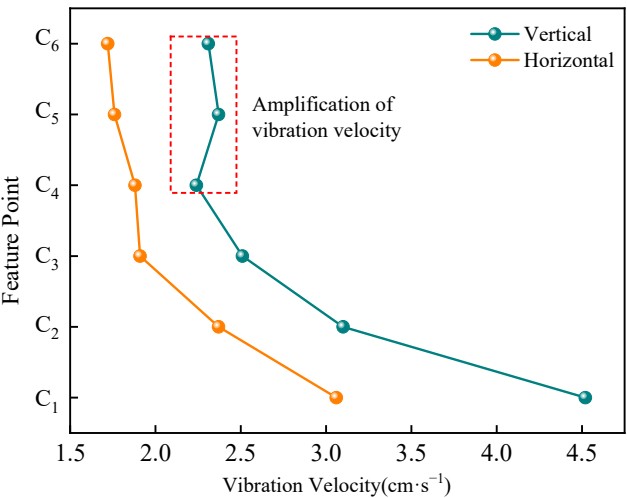

**Figure 5.** Vibration velocities in the vertical and horizontal directions.

### 4.3. Vibration Response Analysis of Slope Blasting

The numerical simulation pictures showed that the vibration velocity was amplified in the small area above the middle of the slope. Figure 6b–d shows the movement of the area where a larger vibration velocity appears. At time 0.004 s, a larger vibration velocity started to appear in the middle of the slope; at 0.005 s, the area was pushed upward; at 0.006 s, the area was at the top of the slope. The movement of the area can be attributed to the slope effect and whip tip effect.

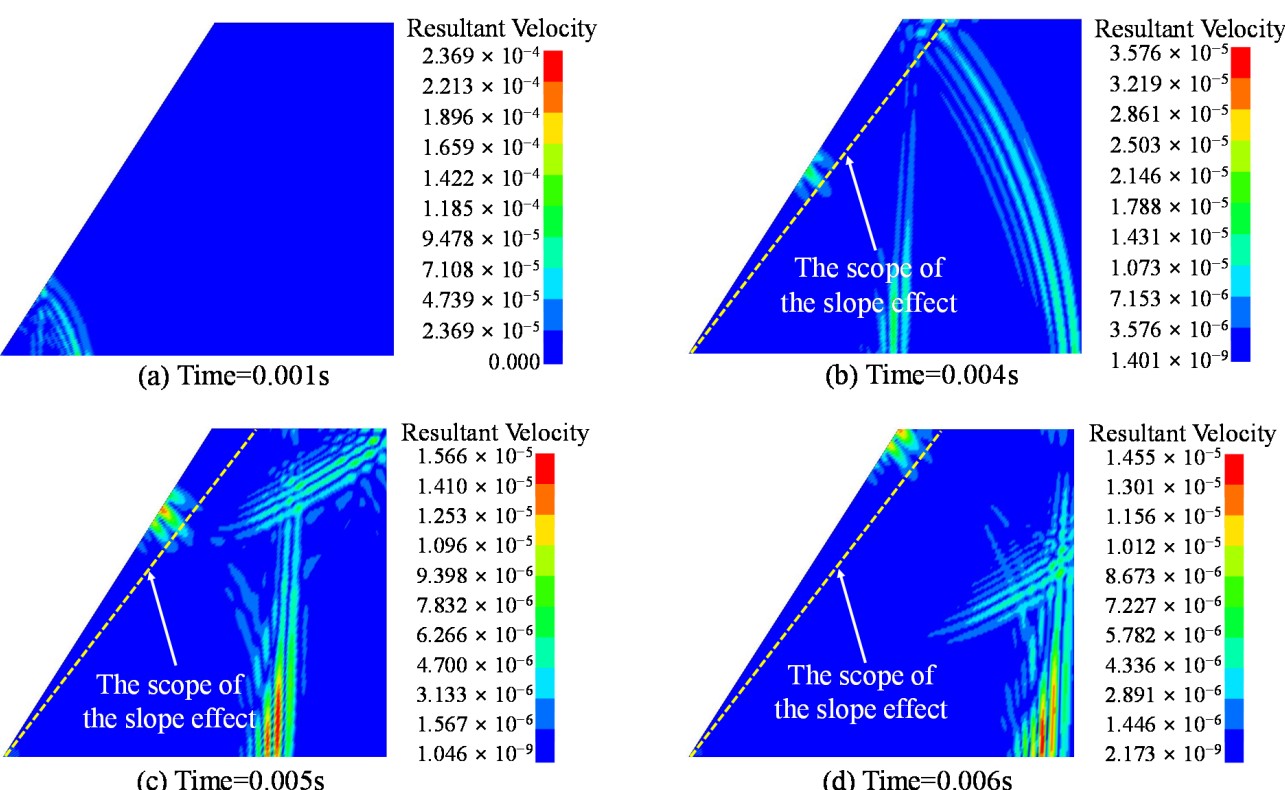

**Figure 6.** Numerical simulation results of blast vibration velocity of the slope.

As the slope foot was close to the explosion center, the energy propagated outward in the form of a shock wave in the area near the explosion. The wave front velocity, pressure, and particle movement velocity of the explosion shock wave were very large, and the attenuation

was not significant. Therefore, the feature points (A1–A6) were set above the middle of the slope in order to better study the blasting vibration changes, as shown in Figure 7.

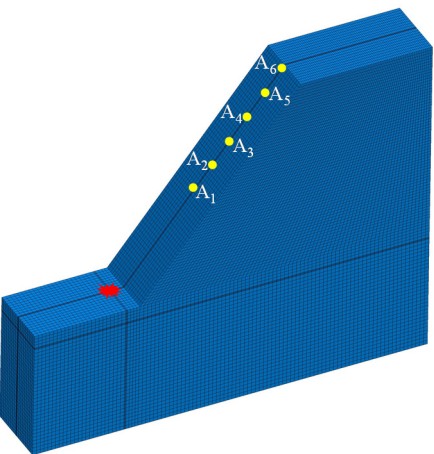

**Figure 7.** The feature points $A_1$–$A_6$.

In this section, the resultant velocity (peak particle velocity, PPV) data were analyzed. The distribution results of slope blasting resultant velocity under four working conditions are shown in Figure 8. It can be seen that in the propagation process of blasting vibration, the resultant velocity was affected by both the attenuation effect and amplification effect, which had their respective dominant scopes.

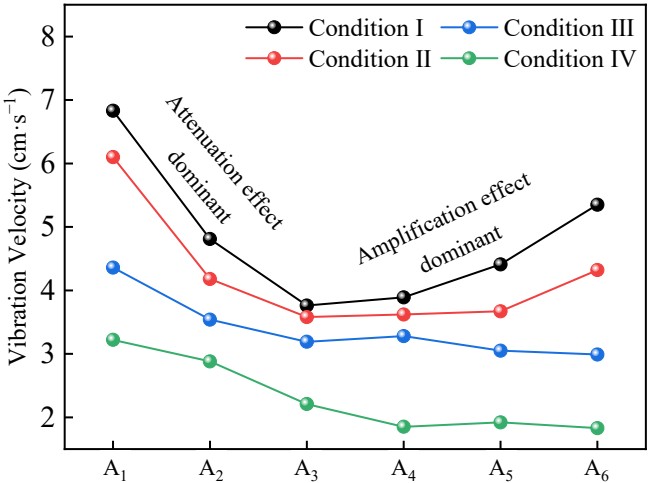

**Figure 8.** The vibration variation curve of the slope.

With the increase of the slope, the vibration velocity of the slope also increased, and the blasting vibration velocity of the slope presented different degrees of attenuation. Within the range from feature point $A_3$ to $A_6$, there was a small degree of amplification under working conditions I and II, with the velocity increasing from 3.89 cm/s and 3.62 cm/s to 5.35 cm/s and 4.32 cm/s, respectively. The blasting vibration velocity increased by 37.5% and 19.3%, respectively. The attenuation effect was weakened and the whipping effect was intensified. Vibration velocity amplification was more likely to occur in the area close to the top of the slope than in the middle part of the slope. Under working conditions III and IV, the slope was relatively small, and under the influence of surface stiffness and stability, the vibration velocity attenuation of $B_4$–$B_6$ point slowed down, and the vibration velocity amplification was not significant.

The amplification rate data were obtained from feature point $A_3$ to point $A_6$ since the amplification phenomenon was mostly found there, as shown in Figure 9. The attenuation

rate data were collected from the characteristic points $A_1$ and $A_6$, considering that the amplification effect had a very limited range, while the attenuation effect was always present. As far as the slope surface is concerned, the amplification phenomenon occurred locally, but overall, the blasting vibration showed an attenuation trend.

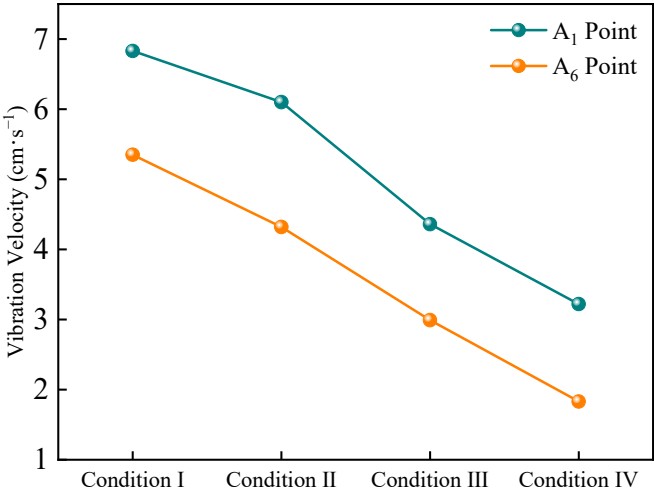

**Figure 9.** The peak vibration velocity decay curve of the $A_1$ and $A_6$ points.

In Figure 9, with the decrease of the slope angle and the increase of the horizontal distance ($D$), the vibration speed at the middle and the top of the slope gradually declined under working conditions I–IV. This phenomenon can be attributed to the slope effect. The area near the bottom of the slope belonged to the topographic abrupt change zone. When the blasting vibration propagates to this area, it will generate a bypass shot and then form a new vibration source. At the same time, when the blasting vibration propagates to the slope surface, it will cause the slope surface to produce a dynamic response. Under the combined effect, the vibration velocity at the slope surface showed an amplification trend [16].

It can be learned from Figure 10 that the attenuation rates of the four working conditions were 21.67%, 29.18%, 31.42%, and 43.16%, respectively, and the changing trend of the vibration velocity attenuation rate was opposite to that of the slope angle. When the slope angle becomes smaller, the attenuation rate of vibration velocity becomes larger, and the elevation effect becomes less significant. Therefore, it can be inferred that when the slope angle is 0, a flat surface will be formed, and the change of blast vibration velocity will not be affected by the height difference. When the slope angle is 90°, the slope surface is perpendicular to the bottom surface of the pit, and a greater vibration velocity will occur at the top of the slope. Due to the limited research content, only the trend of slope vibration velocity variation at 0–90° can be inferred. In terms of amplification, a significant amplification of 29.71% was observed in working condition I, indicating that the slope effect increases with the increase of slope angle. The amplification rates under working conditions III and IV were −6.68% and −20.76%, respectively, and the vibration speed at the characteristic points $A_1$–$A_6$ showed an attenuation trend, but the attenuation was not drastic. This phenomenon can be explained by the whipping effect. When the terrain is flat and homogeneous, the stress state is the same all over the stratum, and there is no big difference in the stiffness and stability of the geotechnical body. The slope area can be interpreted as the protruding part of the stratum, which was subject to weaker constraints and loads than the flat stratum and was less stable [15]. Therefore, compared with the flat stratum, the slope area was more likely to have a larger vibration velocity.

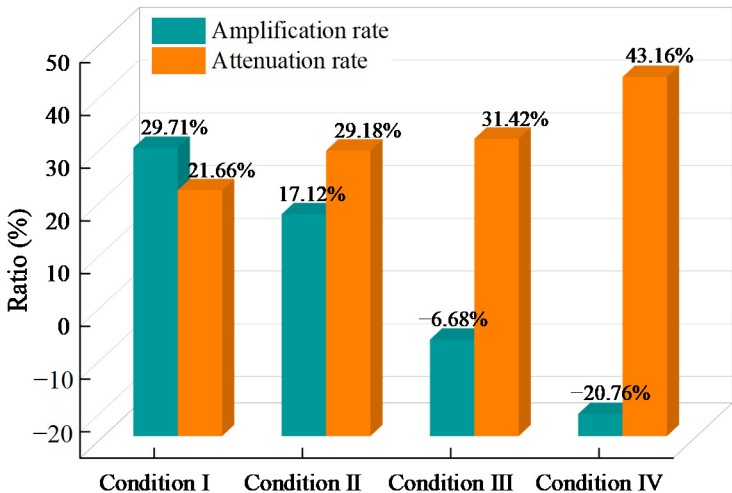

**Figure 10.** Comparison of amplification rate and attenuation rate.

*4.4. Analysis of Variation Law of Internal Vibration Velocity of Slope with Same Elevation*

In order to study the variation rule of vibration velocity (resultant velocity) within the range of 10 m inside the slope, six feature points, namely $B_1$–$B_6$, were set up. Their locations are shown in Figure 11.

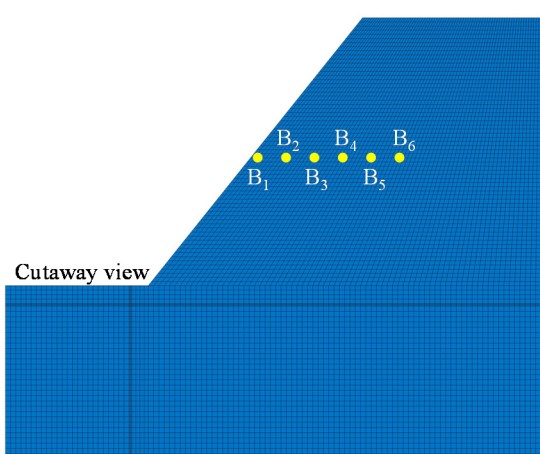

**Figure 11.** The feature points $B_1$–$B_6$.

In this section, the resultant velocity (PPV) data were analyzed. The peak vibration velocity (resultant velocity) of each monitoring point under the four working conditions is shown in Figure 12.

As can be seen from Figure 12, with the increase of horizontal distance (*D*), the attenuation effect was intensified. The peak particle velocities under conditions I–IV were attenuated from 6.83 cm/s, 6.10 cm/s, 4.36 cm/s, and 3.22 cm/s to 2.83 cm/s, 2.30 cm/s, 1.21 cm/s, and 0.86 cm/s, respectively, without an obvious amplification trend. The reasons are as follows. Firstly, compared with the top of the slope, the middle position of the slope had better stability and larger stiffness, where the influence of load and restraint was significantly stronger. Secondly, the increasing horizontal distance was accompanied by the assumption of energy by the blast vibration wave propagation, and therefore the vibration velocity kept attenuating.

The attenuation rate was calculated by extracting the PPV of characteristic points $B_1$ and $B_6$ under working conditions I–IV. The attenuation rate statistics are shown in Figure 13.

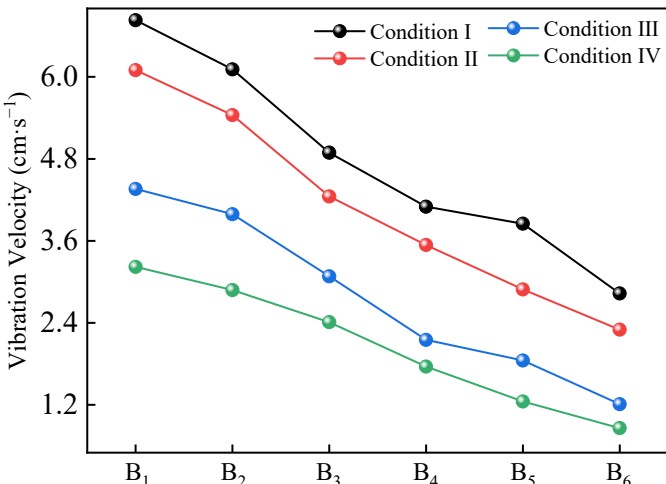

**Figure 12.** The peak vibration velocity of points $B_1$–$B_6$.

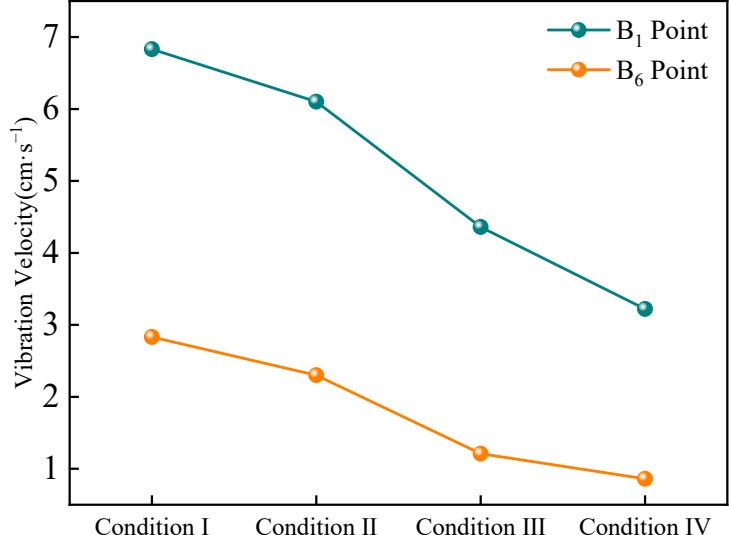

**Figure 13.** The peak vibration velocity decay rate of points $B_1$ and $B_6$.

Figures 9 and 13 show similar regularity. The attenuation rate in Figure 14 is much larger than that shown in Figure 10 because of the different slope effects on feature points $B_1$ and $B_6$. In the interior of the slope, the influence of the slope effect on the blast vibration velocity was not significant. In Figure 12, the decay rate under condition I is the smallest (58.56%) and the decay rate under condition IV is slightly higher (73.29%) since different diffracted waves were generated due to various slope angles. There was also a difference in the range of the slope effect due to the diverse shapes of the slope structure.

Slopes are the key areas to be protected in blasting projects. Therefore, the control and prediction of blasting vibration are critical to the safety of surrounding buildings and the stability of the slope. Blasting vibration field tests should be performed before the blasting. The specifications of explosives to be used for the blasting should be determined in advance, and the site safety work should be performed. With the acquired test data, data fitting was performed to obtain the blast vibration prediction formula. An accurate blast vibration prediction formula can provide a valuable reference for the blasting program. Sadov's formula should be used for blast vibration prediction in flat terrain areas. For the prediction of blast vibration in the slope area, Equation (1) should be used. To comprehensively study the effect of blast vibration on the slope, field monitoring needs to be combined with numerical simulation. With numerical simulation, the peak mass vibration velocity, mass displacement, and unit stress state in the slope area can be obtained. The conclusion data

obtained need to be evaluated based on relevant engineering specifications and existing research to further determine the degree of slope stability and the influence of vibration velocity. However, the numerical simulation method has certain limitations. In actual strata, there are often various complex geological bodies, faults, fracture zones, etc., and therefore, the propagation and attenuation law of blasting vibration propagation are more complicated. Numerical simulation results may be inaccurate for areas with complicated geological conditions.

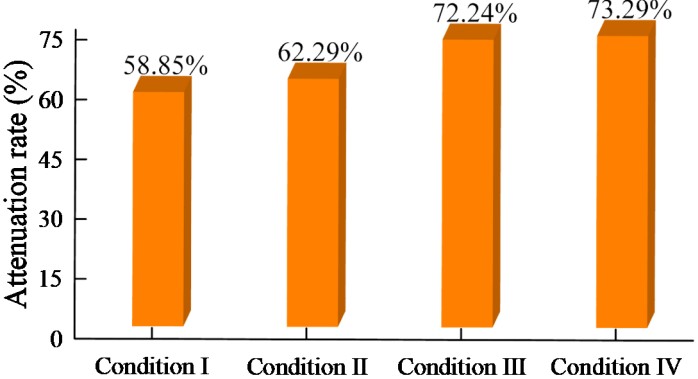

**Figure 14.** Blast velocity attenuation rate of feature points $B_1$–$B_6$.

## 5. Discussion

The object of this paper was to study the blast vibration velocity in the slope area, including the vertical vibration velocity ($V_V$), the horizontal vibration velocity ($V_L$), and the resultant velocity. The analysis of the vibration velocity distribution law proved that the whip tip effect and slope effect occur in the slope area. Many reasonable methods have been developed to study the amplification effect of the internal vibration velocity on the slope. For example, Zhang et al. [29] combined the characteristics of blasting vibration frequency bands and PPV inside the slope to demonstrate the amplification effect of vibration velocity inside the slope. Yan et al. [30] studied the amplification effect of vibration velocity inside the slope by means of field investigation. Torres et al. [31] evaluated the effect of the blasting vibration speed on the slope by means of field monitoring. Tuckey et al. [32] investigated the stability of the slope by remote-sensing technology and mapping, and the same methods were used in the present paper. The study of Tuckey et al. and our research both demonstrated the existence of the whip tip effect inside the slope. However, the sample size in the present study was small, and more samples should be collected to improve the reliability of the conclusions. Jiang et al. [9], Roslan et al. [13], Liang et al. [33], and Huang et al. [34] visualized the amplification effect of blast vibration in the slope by means of numerical simulation. However, numerical simulations cannot fully reflect the conditions inside the strata. Further comparative studies should be conducted and the findings should be verified by field monitoring or theoretical analysis.

## 6. Conclusions

In this paper, two prediction formulas of blasting vibration were modified by multiple linear regression analysis of the measured blasting vibration data. Four working conditions were set for numerical simulation. The blasting vibration response law of slopes with different elevations and the blasting vibration change law of slopes with the same elevation were obtained, and the following conclusions were drawn.

The blasting vibration prediction formula derived in the present study can be well applied to the prediction and control of the blasting vibration at the slope. By comparing Equations (2) with (3), we can infer that the vertical vibration speed was more significantly affected than the horizontal vibration velocity and that the blasting vibration amplification effect was more significant than the attenuation effect. It is worth noting that Equation (1)

is not applicable to blast vibration prediction in flat areas. For blast vibration prediction in flat areas, Sadov's formula should be used.

By analyzing the numerical simulation results, we obtained three conclusions. First, when the horizontal distance (*D*) is constant and the height difference (*H*) becomes larger, the blast vibration in the vertical and horizontal directions shows an overall attenuation trend. The vertical vibration velocity shows a slight amplification at the position near the top of the slope. Secondly, the change of the blasting vibration speed on the slope surface shows the same regularity. Thirdly, the internal vibration velocity of the slope as a whole shows a decreasing trend from the center to the far side. The above three conclusions can be fully explained by the whipping effect and slope effect, and they verify that the whipping effect and slope effect do affect the change of blast vibration inside the slope.

**Author Contributions:** Conceptualization, B.Y.; literature search, B.Y.; investigation, B.Y.; experiment, B.Y.; methodology, B.Y.; data analysis, B.Y.; validation, B.Y., Y.L. and S.D.; formal analysis, Y.L.; resources, M.L.; Q.M. and T.L.; writing—original draft preparation, B.Y.; writing—review and editing, M.L.; figures, B.Y.; supervision, M.L.; project administration, Y.L.; funding acquisition, T.L. All authors have read and agreed to the published version of the manuscript.

**Funding:** Funded by the National Natural Science Foundation of China (Nos. U2006213, U1806230), the Fundamental Research Funds for the Central Universities (201962011), and the Open Foundation of Key Laboratory of Marine Environment and Ecology, Ministry of Education (MGQNLM-KF201804).

**Institutional Review Board Statement:** Not applicable.

**Informed Consent Statement:** Not applicable.

**Acknowledgments:** The authors would like to thank Dong Yan and Chengrong Qing for their help during the experiment.

**Conflicts of Interest:** The authors declare no conflict of interest.

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
