# Peer review of "Study on the Vibration Variation of Rock Slope Based on Numerical Simulation and Fitting Analysis"

_applsci, doi:10.3390/app12094208_

Round 1

Reviewer 1 Report

The paper highlights the variation of vibration velocities  [although it has not been  qualified that which  vibration velocity is referred (ppv or avge pv or the pv?) to in the paper] vis-a-vis the height (vertical distance) horizontal distance and the angularity of slope through field data collection, statistical analysis for regression coefficient and then by developing an ANSYS LS-Dyna based  based numerical models for simulation of different slope angles and evaluate its effect on the attenuation and amplification factors for blasting vibration.

The research idea is good, experimental design is average but the infereneces that are drawn are very poor and cannot be understood clearly. 

Whipping and sloping effects have not been explained clearly through neat illustrations. the Effect of slope material has not been referred to and explained clearly. The results of amplification  and attenuation factors are not clearly described for different slope angles, .

Eqn. 2 is not new eqn, as mentioned by the authors, In fact, the Sadov's Eqn. is site specific and the Eqn. 2 is a simple manifestation of Sadov's Eqn. for the study site.

At any measurement point say "X", situated at a horizontal distance (D) and  elevation difference (H), how the Vv and VL could be segregated at the field scale by seismograph measurements.

The justification for using JWL Eqn. of State to control the process must have been elucidated very clearly vis-a-vis the software usage. 

The loading and boundary conditions of numerical model have not been clearly described.

English language, syntax, grammar and presentation needs serious improvement.

High quality international references are missing that are related or nearly related to the work.

Reviewer 2 Report

This manuscript needs serious revision regarding mathematical symbols used. These symbols are different in equations in style and in paragraph are different. Formatting should be done properly. Tables captions are on other pages whereas tables are on other pages.

Reviewer 3 Report

This paper studied a platform for calculating vibration variation of rock slope based on numerical simulation and fitting analysis. A very preliminary amount of work is carried out. With all the simplifications assumed in the model, the obtained results, make an impression of being an interesting and challenging numerical exercise and thus, unfortunately, diminish the scientific merit of the paper. To minimize these impressions, experimental verification of the obtained results is highly desirable.

The paper cannot be accepted in the present form as it needs further improvements.  

  1. Abstract: The text must be carefully revised. Some sentences contain mistakes. In a research paper, it is expected that the introduction section briefly explains the starting background and, even more important, the originality (novelty) and relevancy of the study is well established. Once this is done, the hypothesis and objectives of the study need to be addressed, as well as a brief justification of the conducted methodology.
  2. The introduction part does not have a flow or direction. It has too many different medical terminologies thrown randomly. Proper references need to be used rather than using others. Language can be improved. The sentences are half-constructed or incomplete so that the readers are expected to fend for themselves to understand their meaning.
  3. Author must be enriching the references with the latest developments in the field. Some of the recent references can be added. The authors have not paid attention to previous research papers and concerns.
  4. The innovation contribution of this article is not clearly stated. The research contributions should be highlighted in the revised manuscript. There is a certain lack of a clear line and message, and my strong advice to the authors would be to consider the overall structure and to either significantly shorten the manuscript.
  5. Provide a proper reference for the equations. It is well known and available in much literature. Please explain and define all the variables in the equations and check the manuscript thoroughly and define the variables where necessary, otherwise, readers cannot understand the equations. 
  6. The convergence of the proposed approach is a critical issue. Please include the convergence study.
  7. There are many linguistic and grammatical typos. please carefully read through and conduct the proofreading. Provide line numbering in the revised version. It is difficult to mention the errors. 
  8. At the end of the manuscript, please describe the scheme of the intended application of the developed method in real practice. What conditions must be met? What preliminary analysis should be carried out? What is the expected performance of this method? What are the limitations of this method?
  9. Discussion Section: Introduce a new section "Discussion", with more current references, which compare the results obtained by the authors with other studies carried out by other researchers. Conclusions Section: Improve the conclusions section, it is very general and does not clearly explain the main objectives achieved in this research.

The list could go on, but the bottom line is that the authors need to rewrite the paper or even reconsider the research content before it could be considered for publication in this journal. 

Round 2

Reviewer 1 Report

Revised version appears to be satisfactory only the presentation needs to be re-checked carefully for  English language, grammar and syntax errors.

Author Response

Thank you for your patience in reviewing. We have made fine touches to the article and corrected the English language and grammatical errors.

Reviewer 3 Report

Comments and recommendations were addressed in the revision. Publication is recommended.

Author Response

Thank you for your patience in reviewing. We will carefully revise the language of the article until it meets the requirements for publication.